# Go Get Data (GGD) is a framework that facilitates reproducible access to genomic data

Michael J. Cormier ⓘ [1,2], Jonathan R. Belyeu ⓘ [1,2], Brent S. Pedersen[1,2], Joseph Brown[1,2], Johannes Köster ⓘ [3] & Aaron R. Quinlan ⓘ [1,2,4✉]

The rapid increase in the amount of genomic data provides researchers with an opportunity to integrate diverse datasets and annotations when addressing a wide range of biological questions. However, genomic datasets are deposited on different platforms and are stored in numerous formats from multiple genome builds, which complicates the task of collecting, annotating, transforming, and integrating data as needed. Here, we developed Go Get Data (GGD) as a fast, reproducible approach to installing standardized data recipes. GGD is available on Github (https://gogetdata.github.io/), is extendable to other data types, and can streamline the complexities typically associated with data integration, saving researchers time and improving research reproducibility.

[1] Department of Human Genetics, University of Utah, Salt Lake City, UT, USA. [2] Utah Center for Genetic Discovery, University of Utah, Salt Lake City, UT, USA. [3] Institute of Human Genetics, University of Duisburg-Essen, Essen, NRW, Germany. [4] Department of Biomedical Informatics, University of Utah, Salt Lake City, UT, USA. ✉email: aquinlan@genetics.utah.edu

There is a need to standardize and simplify access to genomic data to enable reproducibility, remove common barriers to research, and foster studies that integrate diverse datasets. We developed Go Get Data (GGD)[1–5] to address these challenges. Our approach is inspired by software package managers (e.g., pip (https://pip.pypa.io/en/stable/), Conda (https://conda.io), and HomeBrew), which are popular because they use recipes to simplify and automate software installation via standard naming, version tracking, and dependency handling. We realized that the concept of a recipe could also be used to automatically locate, transform, standardize, and install datasets. GGD builds upon the software package framework in Conda, while modifications within GGD allow the Conda infrastructure to support datasets in addition to software. We chose to use Conda because of its wide acceptance and popularity within the life sciences with the support of Bioconda[6], its version tracking and dependency handling, and its ability to normalize the installation of software packages across operating systems. Furthermore, Conda removes the dependence of administrative software management by installing all the desired packages within an isolated environment on a user's system.

Multiple tools and databases have been developed in an attempt to mitigate common problems in accessing genomic data. For example, databases and tools like Galaxy[7,8], the NCBI Assembly Database[9], and Ensembl's set of software resources and APIs[10–12] provide simple access to stable sources of genomic sequence and annotation files. Tools such as the SRA toolkit[13] and Refgenie[14] provide programmatic access to genomic sequencing data. Language-specific tools like AnnotationHub[15] provide access to various genomic data files from resources like UCSC and Ensembl within the R programming language. Additionally, other tools such as Intake (https://intake.readthedocs.io/) provide a means to load multiple data types into data objects for analysis. However, each of these tools or databases are limited in the type and scope of accessible data. Some of these limitations include the quantity and variety of accessible data, the lack of local and/or global data management, the breadth and depth of data availability, and the ability to create and add data recipes to the database. Additionally, some tools are not language-agnostic or do not allow data curation beyond the simple sequence or annotation files. These limitations make existing tools and databases insufficient to provide a reliable resource for genomic data access, reproducibility, and management. GGD attempts to address these limitations by providing a more versatile approach for standardized, reproducible access to genomic data with applications in a wide range of data analyses.

## Results

**GGD data recipe content, creation, and validation**. Conda provides a mature framework on which to build; however, managing genomic data comes with a unique set of challenges not seen with software management. GGD's data recipes require additional knowledge regarding the version and provenance of the datasets, along with details about what makes the recipe unique. For example, genomic data is plagued by many inconsistencies such as genome build, chromosome labeling, sorting, indexing, and more, all of which require consistency and standardization in order to be properly managed. The resulting file format (e.g., BAM[16], VCF[17], BED[18]) must be correct, verified, and standardized for interoperability with common software and other datasets and annotations. Data processing is commonly required in order to use a dataset for an analysis. Processing genomic data also typically requires additional bioinformatics software tools, supplementary genomic datasets, and multiple curation steps. Storing the data recipes, associated metadata,

genomic data files, and more all require a framework for access and management. In order to facilitate the use of data after installation, data files must be consistently organized and have a unique environment variable that points to the specific data package. These difficulties complicate genomic data management and are accounted for within the GGD framework.

Each GGD data recipe is a modified Conda recipe. Conda's recipe format entails a powerful and sometimes complex set of possible statements, only a subset of which is relevant for data recipes. Therefore, in order to simplify the process, GGD partially automates the creation of a data recipe. Rather than requiring researchers to create all of the pieces required for the recipe, GGD only requires one to supply a Bash script (Fig. 1a). The Bash script must contain the necessary steps for obtaining and transforming the raw data into a standardized data recipe. Once a Bash script is provided, GGD command-line tools create (via 'ggd make-recipe') and validate (via 'ggd check-recipe') the recipe for use within the GGD and Conda frameworks.

A GGD recipe contains the information required to find and install the dataset, and to manage the resulting data recipe on a user's system. Each GGD recipe contains a metadata file, a system processing script, a data curation script, and a checksum file (Fig. 1a). The metadata file describes data package information such as software and data dependencies. It also tracks essential attributes such as the species, genome build, data provider, data version, and genomic file type. The system processing script provides recipe, metadata, and local file handling within the Conda environment, initiates data curation, and adds local environment variables for easy data file access. The data curation script provides the necessary steps to access, download, process, and install the data recipe. Finally, the checksum file is used to verify that the data files along with their content are installed as expected. Collectively, these files represent the instruction manual that enables GGD to automatically find, transform, install, and manage genomic data within a local Conda environment on a user's system.

Once a GGD recipe has been created and tested, the recipe is added to the GGD data recipe repository on GitHub. A continuous integration system is used to automatically test and package recipes (Supplemental Note). Once packaged, the continuous integration system caches the resulting data files on cloud storage (currently AWS S3) for rapid user installation, uploads the packaged data recipe as a data package to the Anaconda cloud, and creates the metadata files necessary for use with GGD. This continuous integration system ensures the validity of each recipe and provides an automated approach that simplifies manual review of each data recipe added to GGD.

**GGD data package installation, use, and management**. To install a data package, a researcher using GGD searches for a dataset or annotation by name and/or keyword (Fig. 1b). Once the relevant data package is identified, the researcher uses GGD to install it and integrate it into their research (Fig. 1c). One widely known disadvantage of Conda is the time required to identify all of the dependencies that a software package needs prior to installation (which has, however, been significantly improved in recent releases). Therefore, GGD has optimized the process of installing pre-computed data recipes by bypassing Conda's environment solving step. Instead, GGD directly installs pre-validated data recipes that have been cached on cloud storage, allowing fast data recipe installation (typically in 10 s or less).

Rapid, standardized installation of datasets and annotations removes many common frustrations that researchers face for common analyses. In turn, this increased simplicity allows one to quickly leverage multiple data packages and combine them with

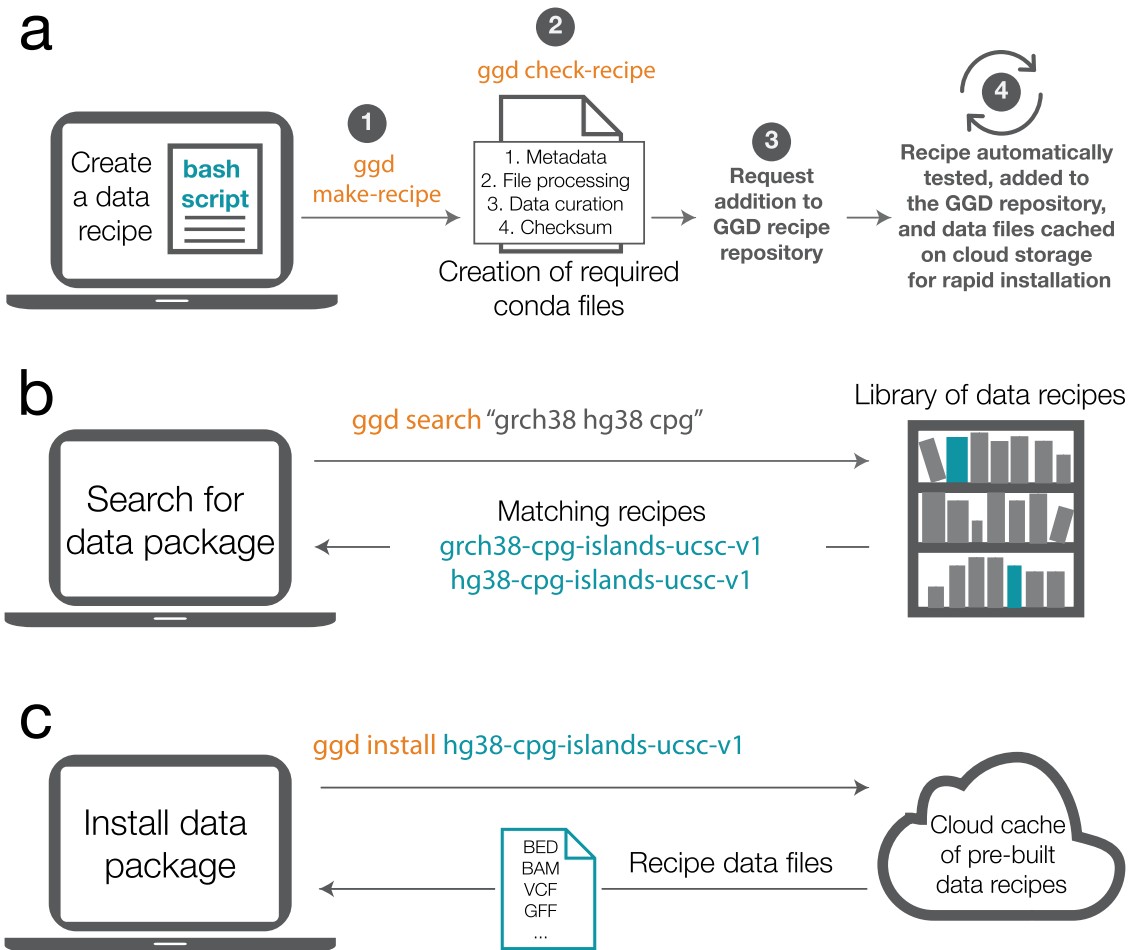

**Fig. 1 Overview of the creation and use of GGD data recipes. a** GGD creates a data recipe from a Bash script, which defines the steps taken to access, process, and curate the desired data files. (1) The "ggd make-recipe" command incorporates the Bash script and additional auto-generated files into a complete data recipe. (2) The "ggd check-recipe" command executes required tests and validates the created data recipe. (3) Once a data recipe has been tested and validated, it can be added to the GGD data recipe repository on GitHub. (4) Each data recipe is further tested via an automatic continuous integration system. If validated, the recipe is transitioned into a data package, which is added to the Anaconda Cloud and the resulting data files are cached on AWS storage. **b** Validated data packages can be found via the GGD command-line interface. For example, to find all data packages associated with "grch38" or "hg38" and the keyword "cpg" one would use "ggd search" with "grch38", "hg38", and "cpg" as search terms. GGD will identify and return all data packages within the GGD library that are associated with the search terms provided. **c** The desired data package is installed via the "ggd install" command. If the data files are cached, they are downloaded directly. If the data package must be built from the recipe, GGD follows the instructions within the recipe while accounting for both software and data dependencies. Installation ends with tracking the version of the installed data package and the creation of local environment variables that facilitate the use of installed data packages. GGD commands are in orange, GGD data packages are in blue.

analysis software to address research questions. For example, Fig. 2 provides a simple use-case in which a researcher can quickly download and integrate necessary GGD data packages into their analyses. In this case, the user installs a GTF file containing coding exons defined by Ensembl[19], as well as a FASTA[20] file for build 38 of the human reference genome. Bedtools[21,22] is then used to identifying the DNA sequences from non-coding loci, which can then be used for downstream analyses. Furthermore, GGD can easily be integrated into reproducible processing and analysis workflow systems such as Nextflow[23] and Snakemake[24].

Information about where the data originated, the version of the data being used, how it was processed, and distinguishing metadata (e.g., genome build for genomic recipes) are key components for every data recipe. This information acts as a unique identifier for the data recipe, and ensures data provenance and reproducibility. GGD maintains this information for each data recipe and provides multiple ways to obtain it through the

documentation page for the recipe, the recipe stored in the ggd-recipes repo, or the ('ggd pkg-info') command (Table 1).

Large-scale data integration is essential in all areas of genome research, since new annotations, datasets, and file formats are constantly being released. Through a suite of command-line tools (Table 1), GGD provides a standardized system for quickly finding, installing, managing, and creating data recipes.

## Discussion

GGD is a natural solution for enabling programmatic access to data in both ad hoc analyses and in more involved, frequently-used workflows. Developed to overcome common problems in genomic data access and processing, GGD provides reproducible and simple access to datasets. Using Conda's version tracking and dependency handling, along with Conda's environment infrastructure, GGD can provide a full range of data management on a user's system. Whether within a container or on a local system,

Data recipe installation creates environment variables to enable simple, reproducible access to the underlying data files

**a**
```
bedtools complement -i $ggd_grch38_coding_exons_ensembl_v1_file \
  -g $ggd_grch38_reference_genome_ensembl_v1_file.fai \
   > grch38-not-coding-exons.bed

bedtools getfasta -fi $ggd_grch38_reference_genome_ensembl_v1_file \
  -bed grch38-not-coding-exons.bed \
   > grch38-not-coding-exons.fa
```

**b**
```
coding_exons=$(ggd get-files grch38-coding-exons-ensembl-v1 --pattern "*gtf.gz")
ref_fasta=$(ggd get-files grch38-reference-genome-ensembl-v1 --pattern "*fa")
ref_fai=$(ggd get-files grch38-reference-genome-ensembl-v1 --pattern "*fai")

bedtools complement -i $coding_exons \
  -g $ref_fai \
   > grch38-not-coding-exons.bed

bedtools getfasta -fi $ref_fasta \
  -bed grch38-not-coding-exons.bed \
   > grch38-not-coding-exons.fa
```

Use the ggd get-files command to retrieve the paths of installed GGD files into environment variables for use across different conda environments

**Fig. 2 Using GGD data packages. a** Data recipe environment variables allow one to use the installed data files without needing to know where the files are stored or how to get them. For example, if one installed the grch38-coding-exons-ensembl-v1 and grch38-reference-genome-ensembl-v1 data packages, one could identify the complement between coding exons and a reference genome using each data file's unique environment variable with the "bedtools complement" command. These environment variables allow one to perform any number of analyses with different bioinformatic tools or scripts. **b** Using the "get-files" command, one can perform the same analysis on coding exons as seen in panel a. With data package environment variables, one needs to be in the environment where the packages were installed in order to use them. Alternatively, the "get-files" command provides access to data files installed by GGD and stored in either the currently active conda environment or a different non-active conda environment. Accessing data files in different environments is supported by the "--prefix" argument. This allows a user to install and store all data packages in a single conda environment while being able to access them from any other environment where GGD is installed. GGD commands are in orange, environment variables that refer to GGD data package files are in blue.

**Table 1 A catalog of GGD tools available via the command-line interface.**

| GGD CLI tool | Description of functionality |
|---|---|
| ggd search | Search for available data packages based on a search term(s) with genomic specific filters. |
| ggd predict-path | Predict the installed file path for a file in a data package that has not been installed. (Useful for workflows like Snakemake). |
| ggd install | Install one or more data package(s) to a specific Conda environment on a user's system. |
| ggd uninstall | Uninstall a data package from a Conda environment from a user's system. |
| ggd list | Report the installed data packages within a specific Conda environment (similar to conda list). |
| ggd get-files | List the file(s) associated with an installed package on a user's system. |
| ggd pkg-info | Retrieve the data package information of an installed data package on a user's system. |
| ggd show-env | Display the GGD specific environment variables for the installed packages in a specific Conda environment on a user's system. |
| ggd make-recipe | Make a GGD data recipe that can be added to the GGD recipe ecosystem. The tool will transform a simple Bash script into a ggd data recipe. |
| ggd make-meta-recipe | Make a GGD data meta-recipe that can be added to the GGD recipe ecosystem. The tool will transform a single or group of scripts into a GGD data meta-recipe, which can be used to install ID specific data packages. |
| ggd check-recipe | Transform a GGD recipe that has been created from running 'make-recipe' into a GGD data package and test the validity of the data package. |

GGD can be used to install data packages before, during, or after the workflow process starts. Environment variables specific to the installed GGD data packages or use of the GGD command-line interface can be used to access the data files for the desired process, including within a workflow. Whether used within a workflow or on their own, GGD data packages provide a simple, reproducible solution to genomic data access, curation, and use.

GGD is actively maintained in order to provide improved functionality and increased access to genomic data. Additionally, the maintenance of Conda by their core development team will provide additional support for GGD. We plan to expand the library of available data recipes within and across species in the short term. This expansion will include recipes for expression data, proteomic data, and many other data types commonly used in genomics. Data recipe development will also be influenced by user feedback and requests.

While currently focused on genomic datasets, the GGD framework has the capacity to support data management across many scientific disciplines. Future development of GGD will include the expansion of GGD data recipes into other non-genomic scientific disciplines. We expect that GGD will help to establish a standard, community-driven ecosystem for reproducible access to genomic

and other scientific data. We encourage contributions from researchers to provide a comprehensive collection of reproducible data recipes to the scientific community. Further information about GGD can be found in the GGD documentation available at https://gogetdata.github.io/.

## Methods

**GGD data recipes**. Data recipes contain the relevant information needed to install and manage data on a user's system. Data recipes are built on the framework of Conda recipes. Conda recipes usually contain a metadata file with information about the software being installed and a script with commands to install that software. The metadata file describes the software, the authors, the programing language, the version, the software dependencies required for building and installing the software, etc. The installation script directs where and how to install the software. Conda uses both of these files to prepare and install the software on a user's system. Conda also has strict regulations on how these files are formatted and what content is provided within these files.

We adapted the Conda recipes framework for use with datasets instead of software. To do this, we worked with Conda and Bioconda to change the formatting requirements so that Conda recipes would work with the additional information requirements needed for data management. This update allowed us to incorporate information such as the data provider, data version, data type, genomic coordinate system, applicable species and genome build. Additionally, it allowed us to incorporate both data and software dependencies within the information file, allowing Conda to provide data and software dependency handling for GGD data recipes. This was particularly important because data recipes commonly go through many data curation steps that require different software packages available through Conda, as well as other data packages available through GGD to be used during that process. Thus, GGD could rely on the mature framework of Conda to provide the correct data and software for data curation without having to set up a dependency handling framework. When creating a GGD data recipe, the user will give data-specific information using the available input parameters in the 'ggd make-recipe' command, which will fill out the data-specific information within this information file.

In addition to the updated data-specific information file, GGD data recipes contain three other essential files. Similar to Conda's software installation script, GGD uses a data curation and installation script. This Bash script is created by the user and contains the required instructions describing where to access and install the data, as well as data curation and data clean-up steps. GGD creates an additional file that is used for data installation and management within a specific Conda environment. Specifically, this script controls the installation path within the user-defined Conda environment, controls required system-level GGD environment variables, initiates data curation from the Conda installed GGD data package, creates specific environment variables for the final data files, and performs other required tasks for data management. The final file contains md5sum hash values used to validate installed data files' content to ensure proper installation.

These four files represent the modified Conda recipes used for data installation and management by GGD. To reduce the amount of work for researchers creating data recipes, GGD requires a user to provide only a Bash script with the necessary data access and curation steps. Using the Bash script and input parameters to 'ggd make-recipe', GGD will create the three additional files that comprise a complete GGD data recipe.

**Using the Conda framework**. Conda (https://conda.io) is a popular software management system that provides version tracking, dependency handling, and environment control on a user's system for many software packages with a wide range of software languages. Conda also removes administrative control over available software, allowing users to access and control needed software on their system. The user-level control of software is defined within a Conda environment, which is maintained and controlled by Conda. These Conda environments provide security and stability for software access and reduce the possibly nefarious results of installing software on a system outside of a controlled environment. Additionally, Conda has been widely adopted within the life sciences community through Bioconda. Because of the popularity of Conda within the life sciences community and Conda's mature management framework, we adapted Conda to provide management for data recipes.

In addition to using modified Conda recipes for GGD data recipes, we use the Conda environment infrastructure to install and maintain data recipes within specific Conda environments, facilitated through Conda's software framework. GGD utilizes most of Conda's internal 'core' functionalities to access and manage data packages. We convert GGD data recipes to data packages using Conda's internal 'build' functionality. This process ensures that the recipe is appropriately formatted for use by Conda, and creates a Conda-usable data package. GGD data packages created in this manner are uploaded and stored in the Anaconda cloud using Conda's 'anaconda' functionality. These packages are then installed into a specific Conda environment through Conda's internal 'install' functionality. The Conda environment is based on Conda's internal 'context', 'envs_manager', and 'prefix' functionalities. Managing installed data packages within a conda environment is supported by Conda's internal 'context', 'list', and 'prefix'

functionalities. Metadata for data recipes are supported by Conda as 'repodata' and Conda core 'index' functionality and are available on the Anaconda cloud under specific GGD Conda channels. Utilizing the Conda framework within GGD allows us to harness the tested and mature infrastructure of Conda without needing to develop a separate one. It also allows for continued maintenance by the Conda core development team, reducing the amount of work required to maintain and improve GGD from the GoGetData development team.

**GGD data management**. The GGD command-line interface (CLI) allows for the access and management of GGD data packages. Table 1 gives a brief overview of the available commands. The GGD CLI uses the Conda software framework, mentioned above, for data management. However, Conda's core functionality does not entirely support data management. Therefore, the GGD software framework integrates elements of the existing Conda framework with additional functionality novel to GGD to provide efficient data management within a Conda environment.

For more information about GGD, see the GGD documentation page at https://gogetdata.github.io/

**Reporting summary**. Further information on research design is available in the Nature Research Reporting Summary linked to this article.

## Data availability
GGD is a data management system developed to help provide sustainable, accurate, reproducible data. No data was used in this manuscript, but data hosted by GGD is available at https://github.com/gogetdata/ggd-recipes[2], and through the GGD command-line interface https://github.com/gogetdata/ggd-cli[1]. Additional information can be found on the documentation page at https://gogetdata.github.io/.

## Code availability
All code for GGD is publicly available on the GoGetData GitHub repository: https://github.com/gogetdata[1-5].

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

## Acknowledgements

Research reported in this publication was supported by funding to M.J.C. from the National Center for Advancing Translational Sciences of the National Institutes of Health under Award Number UL1TR002538 and TL1TR002540. Research was also supported by the National Institutes of Health (NIH) grants: HG006693 and GM124355 to A.R.Q., as well as Essential Open Source Software funding from the Chan Zuckerberg Institute. The content is solely the responsibility of the authors and does not necessarily represent the official views of the National Institutes of Health.

## Author contributions

A.R.Q. conceived the idea for GGD. M.J.C. developed and maintains GGD, the GGD ecosystem, the GGD infrastructure, the GGD recipes, GGD unit and functional tests, Cloud Services used by GGD, and other GGD assets. M.J.C. developed and maintains the docs for GGD, examples, and other material. M.J.C. also wrote the manuscript. J.R.B., B.S.P., and J.B. helped in the development and testing of GGD and GGD recipes, as well as helped in reviewing the docs. J.K. provided insight into management systems and helpful feedback on GGD. A.R.Q. supervised the work on GGD, helped with recipe development, and reviewed GGD assets and docs. J.R.B., B.S.P., J.B., J.K., and A.R.Q. helped in reviews and suggestions for the manuscript.

## Competing interests

The authors declare no competing interests.
