## [Peer Review File · Nature Communications]

Reviewers' Comments:

Reviewer #1:

Remarks to the Author:

This paper introduces a thoroughly documented command-line tool called Go Get Data (GGD) for frictionless retrieval of datasets according to recipes, extending the infrastructure already provided by the Conda software package manager. In bioinformatics, it's now quite easy to install analysis software and develop and deploy workflows across computing environments, but a persistent bottleneck that has until now remained inadequately addressed has been getting and installing datasets in a way that makes them easy to use reproducibly by software in workflows. The GGD framework represents substantial progress to remove this bottleneck. Much code accompanying papers refers to data in directories that readers do not have access to. GGD sets environment variables to represent data files, so code can now refer to these variables together with GGD recipes to facilitate reproducible analysis. I anticipate making use of GGD in future work and enthusiastically recommend publication of the GGD paper after the following issues are addressed:

1) References in this paper are scanty — just to Bioconda (the 1 next to it should be a superscript, BTW), BAM, VCF, BED, Nextflow, and Snakemake. This is all stuff genomics people bang on about, but GGD is being advertised as general-purpose, accommodating all kinds scientific data. If you're going to say that right in your title, you're going to have to show me you care about nongenomic data, that you've made a good-faith effort to look at what folks in other data-intensive fields like cosmology, meteorology, or health informatics do, and how they can use GGD. (Or simply replace "scientific" with "genomic" in your title). Further, aren't there other tools even in genomics that install datasets? Isn't sratools a primitive, annoying dataset installer? I'd talk about this.

2) I think people just write Bash, not BASH.

3) Data files referenced in a recipe are cached on S3? How large can they be? Who pays for this? Is this sustainable? Am I misunderstanding? Is it that the data files are hosted anywhere, and only metadata are on S3? If the data files are hosted anywhere, how does GGD ensure that data versioning is accurate, since data files hosted outside the GGD ecosystem can be replaced?

Reviewer #2:

Remarks to the Author:

The authors have designed a tool for the installation of standardized data recipes.

I tried the GGD installation on macOS and it was working smoothly without any issue.

I tried to understand how GGD could be useful in the everyday work, but I have problem to find some specific advantage.

I run the Use Case to install grch38 reference genome with out any problem. However, after that I got lost. I do not understand what is the advantage of having such a complicated approach to download a fasta genome instead of simply downloading it with the corresponding GTF from ensembl.

If the reproducibility is simply given by the association between a genomic assembly and the corresponding GTF data, I do not see the need of such complicated approach.

Major:

Reading the paper and the supporting info at <https://gogetdata.github.io/index.html#> I am not convinced at all that GGD is a useful tool.

1) Authors need to better address the advantage of using GGD

2) A larger and comprehensive set of examples has to be added to provide convincing evidences that GGD is really useful. An example of workflow using GGD could help in understanding its advantages.

2) On the tutorial it is indicated that the GGD uses Amazon cloud ecosystem. This is a commercial

service, who is going to pay for the service?

Are the authors able to guarantee a long life to their infrastructure?

How is going to be maintained?

3) In the <https://gogetdata.github.io/using-ggd.html> it is indicated that "GGD will no longer maintain python 2 compatibility". This mean that any time a new version of python ends its shelf life the final users need to reinstall all the datasets they are using? Authors must address the effect on the GGD tool usage of end of shelf life of python versions.

Reviewer #3:

Remarks to the Author:

The authors describe Go Get Data (GGD), a system designed to standardize the installation and management of genomic datasets. It is intended to be analogous to a software package manager such as pip, conda, or Homebrew, but applied to datasets rather than software. It is based on the popular conda environment management system and is used in the same way, as a Unix command-line tool by which users search for, install, and use genome-based (and potentially any type of) data. The novel addition is that datasets are built using 'recipes,' which are Unix Bash shell scripts that specify how to download and preprocess each dataset. The use of a publicly available, standard script promotes the reproducibility of the installation and preprocessing steps. Installation of a dataset includes the setting of environment variables that allow the dataset to be referenced independently of where it is installed on the user's filesystem. Recipes reside on a GitHub repository, and dataset creators are encouraged to submit their own recipes for inclusion.

The application of a package/environment management system to genome datasets attempts to address the significant challenge of how to manage a collection of datasets created using different parameters, e.g., genome build, organism, etc. The system takes a genome-centric view, making it easy to retrieve gene annotation tracks and similar data that is directly linked to coordinates on a given genome build. However this focus limits the applicability of the system to the wider scope of genomic data, including gene expression, proteomics, flow cytometry, and other datatypes that are further removed from genome track type information. Additionally, there is an assumption, inherited from software package managers, that datasets are retrieved via download from a web resource, ignoring the many genomic data repositories that require queries to specify which data should be retrieved. The number of genomic datasets available through this system is therefore a fraction of the datasets in the genomic data space. Finally, the use of the Bash shell as a means to specify recipes is effective but not general enough for the wider population of researchers who are likely not fluent enough in Unix to benefit from their use. For these reasons, the scope of this paper makes it more suitable to a bioinformatics-specific publication than the Nature Communications audience.

The authors provide a system for managing datasets and specify a recipe format by which dataset providers can make their packages available on the GGD repository. This process requires creating a shell script that downloads the dataset and performs transformations such as sorting and filtering to produce the final result. They must make the case that there is some benefit of this process above and beyond simply making the final dataset available for download that justifies the effort for the author of creating and submitting the recipe, and for the user of installing conda and using the ggd system.

One potential benefit is reproducibility - the shell script describes exactly the steps that are taken to produce the dataset, and this is useful for those attempting to understand how the dataset was created. However, a large number of available recipes consist of simply downloading a dataset from an FTP site or providing minor modifications. Other recipes are more complex but rely on processes that are questionable from a production quality standpoint, such as using Python to extract a gene list from an Excel spreadsheet. This is an improvement over doing the same thing by hand due to its transparency, but the understanding of these transformations is limited to those with Unix fluency.

While this expectation is less of a problem in the world of software development, the genome researcher in the street is not comfortable enough parsing Bash scripts. The authors have decided that the Bash script is sufficient to remove the complexity of creating a full Conda recipe, and for most of the available recipes this is the case. However, because of the limitation to Bash scripts, recipes with slightly more complexity have to resort to techniques such as "here" documents to embed and execute external scripts. Again, this is not a problem for an experienced computational biologist, but as a tool that is proposed as a general purpose solution, the authors choice of Bash shell as the format for reproducibility limits its wider acceptance.

Another limitation is the assumption that datasets are available to be downloaded directly. This assumption is true for the type of recipes currently in GGD - tracks comprising genes, variants, or other data specific to a genome build. These require a download of a reference genome, a download or computation of a set of genomic coordinates corresponding to the desired genes, along with their annotations. However, many important datasets reside in repositories that require users to specify a query in order to download data. Examples include GTEX, Sequence Read Archive (SRA), Gene Expression Omnibus (GEO), and Ensembl, as well as disease-specific repositories such as The Cancer Genome Atlas (TCGA) and the Type 2 Diabetes Knowledge Portal, and many others. While many of these repositories make their data available for bulk download, this is not usually the way researchers retrieve it, and therefore it would take significant additional effort to create a recipe for a dataset from a query-based repository.

The authors mention that, when a dataset is installed, an environment variable is automatically created that points to the directory where the dataset is located. In this way, the user does not need to know where the dataset is in order to use it. This is a useful feature, but it is not so much an advantage as a necessity. Because the system is based on conda, datasets are installed within the anaconda distribution filesystem - not a place where an analysis workflow should be run.

Another problem is the limited scope of the search functionality. For example "ggd search variant" returns 35 packages, several of them from the ClinVar database, but "ggd search clinvar" returns no results. In a review of a more standard bioinformatics tool paper, the software details are less important, but the success of GGD will depend critically on its ability to build a community of recipe users and developers, and search functionality is a basic prerequisite for this.

The features of the system are well-described, and because it is based on one of the most popular package managers, its commands and syntax are familiar to those with conda experience. The system runs smoothly and provides helpful feedback. One aspect that the paper could describe in greater detail is how a recipe is validated and what is required for it to pass continuous integration tests.

While this paper is of interest to a more technical audience of bioinformaticians, it describes an approach to the problems of reproducible research and data management that is a novel application of technology, and, if made more accessible to less technical users and compatible with different omics data types, could be a substantial contribution. For this reason I recommend publication in a journal more specific to technical bioinformatics users.

We thank all three reviewers for their detailed review and for the helpful comments and suggestions that have improved the clarity of our manuscript. In the sections that follow, we have responded to each reviewer comment "in line" using blue text.

Reviewer #1 (Expertise: Bioinformatics and community platforms):

This paper introduces a thoroughly documented command-line tool called Go Get Data (GGD) for frictionless retrieval of datasets according to recipes, extending the infrastructure already provided by the Conda software package manager. In bioinformatics, it's now quite easy to install analysis software and develop and deploy workflows across computing environments, but a persistent bottleneck that has until now remained inadequately addressed has been getting and installing datasets in a way that makes them easy to use reproducibly by software in workflows. The GGD framework represents substantial progress to remove this bottleneck. Much code accompanying papers refers to data in directories that readers do not have access to. GGD sets environment variables to represent data files, so code can now refer to these variables together with GGD recipes to facilitate reproducible analysis. I anticipate making use of GGD in future work and enthusiastically recommend publication of the GGD paper after the following issues are addressed:

We would like to thank review number 1 for their review of our manuscript. We appreciate the comments and feedback they provided. We have addressed the issues raised by reviewer 1 in the manuscript and describe them below.

1) References in this paper are scanty — just to Bioconda (the 1 next to it should be a superscript, BTW), BAM, VCF, BED, Nextflow, and Snakemake. This is all stuff genomics people bang on about, but GGD is being advertised as general-purpose, accommodating all kinds scientific data. If you're going to say that right in your title, you're going to have to show me you care about nongenomic data, that you've made a good-faith effort to look at what folks in other data-intensive fields like cosmology, meteorology, or health informatics do, and how they can use GGD. (Or simply replace "scientific" with "genomic" in your title).

The reviewer has rightfully pointed out that our manuscript mentions, especially in the title, that GGD is a general-purpose scientific data management system but there is little content about non-genomic data. We agree that our references and content is focused on genomics, which has been our initial area of focus. Accordingly, we have applied the reviewer's recommended changes to the title, along with changing to comply with formatting guidelines, which now reads "Go Get Data (GGD) provides simple, reproducible access to genomic data" where the changes are highlighted in yellow in the manuscript. At the end of the manuscript, it states "While currently focused on genomics datasets, the GGD framework has the capacity to support data management across many scientific disciplines." We have added an additional sentence stating the intended work to

expand GGD into other scientific disciplines. This sentence has been highlighted in yellow.

Further, aren't there other tools even in genomics that install datasets? Isn't sratools a primitive, annoying dataset installer? I'd talk about this.

Thank you for pointing out that we had too few references, especially for tools that strive to provide similar data access to genomic data. We have added a paragraph, highlighted in yellow in the manuscript, briefly covering a few databases and tools available for genomic data, along with a short description of what those tools lack that GGD is equipped with. We have also updated the references based on these changes.

2) I think people just write Bash, not BASH.

Thank you for pointing out a few minor spelling/formatting mistakes. We have updated the manuscript with the suggested changes and have highlighted those changes in yellow.

- The "1" next to Bioconda has been updated to a superscript and highlighted in yellow.
- "BASH" has been updated to "Bash" and highlighted in yellow.

3) Data files referenced in a recipe are cached on S3? How large can they be? Who pays for this? Is this sustainable? Am I misunderstanding? Is it that the data files are hosted anywhere, and only metadata are on S3? If the data files are hosted anywhere, how does GGD ensure that data versioning is accurate, since data files hosted outside the GGD ecosystem can be replaced?

The reviewer raises valid concerns about the sustainability of caching data packages on Amazon S3. The final processed data from a recipe is stored on AWS S3. Currently, all recipes in GGD are stored on AWS S3. However, those recipes stored on AWS will be limited to the size of the data files and we currently do not cache any file larger than 5 gigabytes. The cost of AWS S3 storage is currently covered by lab funding and further funding for AWS S3 storage and management is being pursued, but as we detail below, there are more sustainable solutions for caching in the event that longer-term funding is infeasible.

First, our rationale for caching in the first place. We host the product of as many data recipes as possible, currently all recipes, on AWS for two purposes. Firstly, to decrease the time it takes to install the data files by bypassing the data processing step. Secondly, to ensure data provenance and accuracy. Some data providers do change the contents of a data file hosted on their site. We ensure data versioning, provenance, and reproducibility by caching the data files created by the recipe on AWS at the time of data recipe acceptance into GGD.

Additionally, md5sum hash values are generated for each data file created by a recipe at the time of recipe creation. These hash values are maintained in the metadata of a recipe and are used to ensure that the contents of a data file have not changed from the time of creation to the

time of installation on a user's system. The combination of AWS caching and md5sum hash value checking ensures accurate, unchanged reproducible data.

In the future, we may need to implement a limit on the size or type of data that is cached on AWS. Approaches we have considered include:

- 1) Setting a cap on the number of files that are cached such that at most N files per M total Gb are cached, all other recipes are installed from scratch
- 2) Add a system that caches the top N most commonly used recipes, updated on a rolling basis.
- 3) One other approach could be to use a different cloud storage system, or host a local site where cached files are stored.

We emphasize that caching is a *convenience*, not a *necessity* of our system. It is provided so that vetted data packages can be installed as quickly as possible. In the worst case, we envision a scenario where only the top N most frequently used recipes are cached and other recipes are built from scratch following the recipe's instructions.

Reviewer #2 (Expertise: Bioinformatics and community platforms):

The authors have designed a tool for the installation of standardized data recipes. I tried the GGD installation on macOS and it was working smoothly without any issue. I tried to understand how GGD could be useful in the everyday work, but I have problem to find some specific advantage. I run the Use Case to install grch38 reference genome with out any problem. However, after that I got lost. I do not understand what is the advantage of having such a complicated approach to download a fasta genome instead of simply downloading it with the corresponding GTF from ensembl. If the reproducibility is simply given by the association between a genomic assembly and the corresponding GTF data, I do not see the need of such complicated approach.

Major:

Reading the paper and the supporting info at <https://gogetdata.github.io/index.html#> I am not convinced at all that GGD is a useful tool.

- 1) Authors need to better address the advantage of using GGD

We regret that the motivation for GGD was unclear, which may be due to a lack of clarity regarding the breadth and depth of possibilities available through GGD. If GGD were merely used for reference genomes, then the reviewer would be correct in that GGD is a complex approach for installing a simple fasta file.

However, GGD currently contains many different types of genomic data files along with the capability of hosting many more. The majority of these data files cannot just be downloaded from a host site due to the required, intermediate data processing, which itself often imposes

other data and software dependencies. The GGD recipes facilitate this often complex process and establish a high level of transparency and reproducibility.

A major advantage of GGD is that it allows recipes to be fitted with a range of simple to complex data curation steps. This means that not only does a recipe provide access to general data files like a fasta file or a gtf annotation file, but it allows and provides for a recipe to be created with multiple steps of data curation using different software and data dependencies. Furthermore, this approach allows a research or a team to utilize customs in house scripts they have developed for their data curation process and provide those steps within a GGD data recipe, ultimately supplying the scientific community with a transparent, tested, and reproducible set of instructions and data files that otherwise may be difficult to obtain, as well as allowing the research or team to simply access, reference, and distribute their data recipes.

Another major advantage of GGD is it allows a user to install data files relevant to their research and experiments without taking the time to find, install, process, and QC the data by hand. Using GGD minimizes many common errors in data analysis due to mistakes made during the data curation process.

GGD also has similar advantages to software package managers. Some of these include simple access to a growing set of processed data files through data packages, package versioning, software and data dependency handling, etc. For example, some recipes may require multiple software and data packages in order to fully process the data. Using the modified recipe scheme in GGD allows for the system to manage those required dependencies for data processing and installation. This means that a user does not need to worry about installing additional software or data prior to obtaining a recipe because the management system handles it for them. The GGD system manages the data files once installed on a user's system much like a software package management system does. It also allows one to create a conda environment where all data files can be installed and stored while being able to access all of those data files from any other environment. This reduces the redundancy of data installation on a system. It also allows a team of researchers using the same file system to access and use the data without each having to install it separately. It allows researchers to catalog the data processing they did for experiments and easily provides those data files as reproducible data recipes in GGD rather than custom scripts on some repository that commonly fails to work.

Furthermore, GGD is set up to work with workflows like Snakemake or Nextflow. This means that someone can create a reproducible workflow that uses a set of GGD recipes. This workflow can be distributed without the need to provide customized data files or additional steps for data processing. These advantages are just a few of the many that come with having a data management system like GGD. These advantages are detailed in the main manuscript.

2) A larger and comprehensive set of examples has to be added to provide convincing evidences that GGD is really useful.

GGD currently hosts over 350 recipes split across different species and species-specific genome builds, with most of these recipes specific to Homo sapiens. Recipes range in complexity of data processing and data type provided. Some examples of available recipes include Recipes for gene features such as general gtf files, canonical isoforms and transcripts, tRNAs, lncRNAs, introns, UTRs, protein-coding genes, coding regions, etc. Recipes for genomic tracks such as segmental duplications, cpg islands, pfam domains, microsatellites, etc. Recipes for conservation and constraint metrics such as pLI, missense z, constrained coding regions, phastcons, evofold, etc. Recipes for reference/sequencing data such as reference fasta files, amino acid sequences, etc. RNA editing datasets such as RADAR. Gene sets such as Autosomal Dominant genes, Autosomal Recessive genes, Haploinsufficient genes, etc. Recipes for variant files such as variants from the Exome Sequencing Project (ESP), clinically associated variants, somatic variants, structural variants, etc. Recipes for chromosome sizes and mapping, including liftover chain files, chromosome sizes by genome build, and mapping files for variations of the same genome build. Additionally, particular types of recipes exist called meta-recipes used to access ID specific recipes, thus making them able to provide extensive access to countless recipes. One such example is the Gene Expression Omnibus (GEO) meta-recipe, which can take any GEO accession ID and provide the relevant data for that specific accession ID.

GGD is equipped to host a vast range of data types with the ability to modify the complexity of data processing for each recipe. This allows GGD to provide an extensive set of data recipes that can fit the needs of researchers. It also allows researchers to create their own data recipes and host them on GGD, thus fostering data provenance, reproducibility, and dissemination.

An example of workflow using GGD could help in understanding its advantages.

Thank you, this is a very helpful suggestion. We have added new examples in the GGD documentation, including examples of using GGD in a Snakemake workflow and a Nextflow workflow. We feel that the current examples in the GGD docs as well as a good understanding of GGD and its capabilities provide evidence for the usefulness of GGD.

2) On the tutorial it is indicated that the GGD uses Amazon cloud ecosystem. This is a commercial service, who is going to pay for the service?

This is a valid concern that was also raised by Reviewer 1. We invite Reviewer 2 to look at our response to Review 1 on the same subject.

Are the authors able to guarantee a long life to their infrastructure?
How is going to be maintained?

GGD has a core team of developers. Just like bioconda, GGD will be maintained by both the core team as well as through community contributions. This is a large reason why bioconda has been so successful. We intend that GGD will be as popular and productive for data as bioconda has been for bioinformatics software.

3) In the <https://gogetdata.github.io/using-ggd.html> it is indicated that "GGD will no longer maintain python 2 compatibility". This means that any time a new version of python ends its shelf life the final users need to reinstall all the datasets they are using? Authors must address the effect on the GGD tool usage of end of shelf life of python versions.

The reviewer identified from the GGD docs that GGD will no longer maintain python2 compatibility. Their understandable question is what the impact of this is on the GGD ecosystem for users. We emphasize that the GGD command line interface (CLI) is the only part of GGD that is dependent on python. The data recipes and installed datasets are not.

The statement found in the docs refers to the fact that the GGD team will no longer maintain python2 compatibility with the GGD CLI because the python core team no longer supports that version of python. This means that in order to use a newer version of GGD one would need to do so in an environment where python ≥ 3 is installed. A simple update of a python version in a conda environment can solve the problem if the user wants to continue to use an environment where python 2 was installed.

Data from GGD data recipes that have been installed in a python2 environment **do not need to be reinstalled**. GGD is still able to access that data from different conda environments. For example, if a user was to store all of their data recipes in a conda environment named "ggd_data" which uses python2, the users could simply use the "--prefix" parameter in GGD to install and access installed data in the "ggd_data" environment from any other environment. Additionally, the python version of that environment can be updated to python3 to work with newer versions of the GGD CLI.

In the case where there is a recipe that is dependent on python 2, then that recipe needs to be updated. For future use. However, if the data from that recipe has already been installed, then the user **does not** need to uninstall it.

We would like to note that the migration from python2 to python3 is not a singular event from GGD, but rather is a global event for all python users. Any tool that uses python2 is working with the same restrictions and same obligations to migrate to the maintained version of python.

Reviewer #3 (Expertise: Bioinformatics and community platforms):

The authors describe Go Get Data (GGD), a system designed to standardize the installation and management of genomic datasets. It is intended to be analogous to a software package manager such as pip, conda, or Homebrew, but applied to datasets

rather than software. It is based on the popular conda environment management system and is used in the same way, as a Unix command-line tool by which users search for, install, and use genome-based (and potentially any type of) data. The novel addition is that datasets are built using 'recipes,' which are Unix Bash shell scripts that specify how to download and preprocess each dataset. The use of a publicly available, standard script promotes the reproducibility of the installation and preprocessing steps. Installation of a dataset includes the setting of environment variables that allow the dataset to be referenced independently of where it is installed on the user's filesystem. Recipes reside on a GitHub repository, and dataset creators are encouraged to submit their own recipes for inclusion.

The application of a package/environment management system to genome datasets attempts to address the significant challenge of how to manage a collection of datasets created using different parameters, e.g., genome build, organism, etc. The system takes a genome-centric view, making it easy to retrieve gene annotation tracks and similar data that is directly linked to coordinates on a given genome build. However this focus limits the applicability of the system to the wider scope of genomic data, including gene expression, proteomics, flow cytometry, and other datatypes that are further removed from genome track type information. Additionally, there is an assumption, inherited from software package managers, that datasets are retrieved via download from a web resource, ignoring the many genomic data repositories that require queries to specify which data should be retrieved. The number of genomic datasets available through this system is therefore a fraction of the datasets in the genomic data space. Finally, the use of the Bash shell as a means to specify recipes is effective but not general enough for the wider population of researchers who are likely not fluent enough in Unix to benefit from their use. For these reasons, the scope of this paper makes it more suitable to a bioinformatics-specific publication than the Nature Communications audience.

The authors provide a system for managing datasets and specify a recipe format by which dataset providers can make their packages available on the GGD repository. This process requires creating a shell script that downloads the dataset and performs transformations such as sorting and filtering to produce the final result. They must make the case that there is some benefit of this process above and beyond simply making the final dataset available for download that justifies the effort for the author of creating and submitting the recipe, and for the user of installing conda and using the ggd system.

In our opinion, the main reasons for supporting and exposing the recipes themselves is reproducibility and transparency with respect to the provenance and modifications that have been made to the data. Without this, it is a black box and will not instill researcher confidence nor allow researchers to inspect the recipes nor improve them over time.

GGD recipes are a reproducible log of where the data was obtained, how it was processed, and what is included. Beyond just the data curation scripts themselves, each

recipe contains metadata that is used to manage the data files on a user system along with information about what the data is, where it is from and important features of the data. Metadata in each recipe also includes md5sum hash values used to validate the content of installed data files. Any software or data dependencies are included in the metadata, and are automatically handled during data package installation.

As a whole, the recipe stands as the instructions and ingredients for obtaining processed, stable, reproducible, and ready to use data. The recipe system has been proven with software package managers and using it for data management allows us to take advantage of the power of the recipe system. Additionally, GGD caches the final data files from a recipe on AWS to reduce installation time and ensure data provenance and reproducibility. Therefore, researchers can confidently install the datasets and annotation required for their experiments and analyses without spending a large amount of time trying to do it by hand, trying to reproduce steps outlined in another manuscript, or trying to use custom scripts on a repository that may or may not work. Additionally, using GGD reduces the errors that occur during data curation and analysis because of the potential by hand processing errors.

Once installed on a users system, GGD provides the tools to manage, access, and use the installed data. GGD should be used to minimize redundant data across a users system by installing data into a user defined dedicated conda environment for data. Data installed in one conda environment can be accessed across any other conda environment where GGD is installed. Additionally, data can be installed in any conda environment on a users system and is not constrained to the currently active environment. This means that a user does not need to worry about where the data is installed within a complex file system, but rather can focus on the analysis at hand while being able to access and use the installed data from GGD. GGD recipes provide a stable, reproducible, and accurate data resource, and the GGD CLI provides a simple tool to access, manage, and use installed data on a users system. Together, the GGD system provides a powerful and easy to use tool to access scientific data relevant to a user's experiments, analyses, and workflows, minimize data curation error, manage that data across complex file systems without needing to know the intricacies of that file system, along with many other major benefits.

One potential benefit is reproducibility - the shell script describes exactly the steps that are taken to produce the dataset, and this is useful for those attempting to understand how the dataset was created. However, a large number of available recipes consist of simply downloading a dataset from an FTP site or providing minor modifications. Other recipes are more complex but rely on processes that are questionable from a production quality standpoint, such as using Python to extract a gene list from an Excel spreadsheet. This is an improvement over doing the same thing by hand due to its transparency, but the understanding of these transformations is limited to those with Unix fluency. While this expectation is less of a problem in the world of software development, the genome

researcher in the street is not comfortable enough parsing Bash scripts. The authors have decided that the Bash script is sufficient to remove the complexity of creating a full Conda recipe, and for most of the available recipes this is the case. However, because of the limitation to Bash scripts, recipes with slightly more complexity have to resort to techniques such as "here" documents to embed and execute external scripts. Again, this is not a problem for an experienced computational biologist, but as a tool that is proposed as a general purpose solution, the authors choice of Bash shell as the format for reproducibility limits its wider acceptance.

We chose Bash because it is relatively easy to use, it has become a necessity to use in genomics, and because of its versatility. If the process is simple and only requires downloading a dataset from an FTP site even with common minor modifications like sorting the data, Bash is a great option. Additionally, any UNIX tool that works in Bash, such as bcftools or bedtools, can be used in a Bash script. This includes the majority of bioinformatics tools. Furthermore, one can write custom processing scripts with Bash or in the programming language of their choice and either embed them in the Bash script or include them in the recipe directory where the Bash script uses that custom script. Bash provides the simplicity and versatility for beginner computational scientists to use as well as the power for experienced computational biologists to harness. In our opinion, a Bash script provides a good balance between usability and versatility. The reviewer states that there are some limitations to Bash scripts, however, we would argue the opposite that it minimizes limitations. It may be true that "a typical genome researcher" may not be well equipped to create a complex data recipe. If this is the case it probably means that they should not be creating complex data recipes or that they should learn the UNIX systems enough to operate Bash. Additionally, researchers who either do not have the skills to create their own data recipe or who would like help have the option to request a data recipe be created by the GGD team. Links exist in GGD GitHub repos as well as on the GGD docs for a user to request a data recipe.

Another limitation is the assumption that datasets are available to be downloaded directly. This assumption is true for the type of recipes currently in GGD - tracks comprising genes, variants, or other data specific to a genome build. These require a download of a reference genome, a download or computation of a set of genomic coordinates corresponding to the desired genes, along with their annotations. However, many important datasets reside in repositories that require users to specify a query in order to download data. Examples include GTEx, Sequence Read Archive (SRA), Gene Expression Omnibus (GEO), and Ensembl, as well as disease-specific repositories such as The Cancer Genome Atlas (TCGA) and the Type 2 Diabetes Knowledge Portal, and many others. While many of these repositories make their data available for bulk download, this is not usually the way researchers retrieve it, and therefore it would take significant additional effort to create a recipe for a dataset from a query-based repository.

The reviewer makes a helpful argument that there are many websites where it is not feasible to download datasets for use in GGD. Their assumption is that it is better to use

the online query system to partition the data they would like to use for an experiment. We argue that this process is inconsistent and not reproducible. It is common for researchers to use online query tools to access data, but they commonly forget the steps that were taken or make mistakes in describing the process of data querying. The different data providers the reviewer mentioned have multiple options for data access and downloading through the command line, including FTP sites and multiple different query options. It is common for data providers to provide a query CLI or other tools to access, query, and filter their datasets at the command line. Although this may be unconventional for “a typical genome researcher”, this is a common approach to access and use data from these sites. We therefore argue that one can just as easily use a query language such as MySQL, if available for the datasets, to query the desired data. Additionally, query languages are easily accessible and usable within Unix systems and can be used in Bash scripts. Again, we feel this is an advantage rather than a limitation. It provides researchers the ability to correctly query and access their desired data as well as provide a reproducible log of what they did and how they did it. Additionally, if the data provider were to update the dataset being queried, using the exact same steps from an online query would likely lead to different results. On the other hand, because of how GGD is set up the data files installed would be exactly the same as they were prior to the data update on the provider site.

One major goal of GGD is to reduce the inconsistencies with data access and use. Again, it may be common to use online query tools but there are major obstacles that commonly cause inconsistency and non-reproducible results. Although it may be difficult for some researchers to transition, we believe that the additional effort is worth the results.

The reviewer also mentions their concern on how GGD is limited because it is set up to only work with “track” data types. Although the majority of data recipes in GGD do follow the “track” data type, GGD is not limited to only “track” type data. GGD has been set up to work with many different types of data, with an easily modifiable framework for data type integration. The structure of GGD is used to determine and propagate data, and that structure is accommodating to multiple different types of data, including non “track” type data. This allows GGD the versatility to provide many different types of data with the ability to maintain internal structure so that minimal changes are needed for updates to data type and data type testing.

For example, based on Reviewer 3’s input, we have created a recipe for the Gene Expression Omnibus (GEO). This type of recipe is considered a meta-recipe, as it allows a user to use an GEO accession identifier to install data specific to that GEO accession ID. For the GEO meta-recipe, there is no limitation on the type of expression data that can be downloaded, the only limit is based on the available data files for that accession ID. This meta-recipe is available and active in the GGD ecosystem and can be downloaded using the latest version of GGD. We have also added additional documentation about meta-recipes in the GGD docs along with an example of using this GEO meta-recipe in a

SnakeMake workflow. Meta-recipes and GEO stand as one example of the versatility of GGD and its ability to adapt and customized to the vast array of database and data types.

The authors mention that, when a dataset is installed, an environment variable is automatically created that points to the directory where the dataset is located. In this way, the user does not need to know where the dataset is in order to use it. This is a useful feature, but it is not so much an advantage as a necessity. Because the system is based on conda, datasets are installed within the anaconda distribution filesystem - not a place where an analysis workflow should be run.

The reviewer is correct in that the data files are installed within the conda filesystem. This allows GGD to manage data recipes and associated data files within and across different conda environments. GGD provides a tool called "get-files," which is used to get the file paths for the various data files installed by GGD. The "get-files" command fills the necessity of data access within the conda filesystem and the ability to utilizing data recipes for analysis workflows, all without needing knowledge of the conda file system. In addition to the "get-files" command, GGD creates directory and data file specific environment variables used to access the installed data within a particular environment of conda. These environment variables provide additional "self-documentation" for the recipe's data files, and allow an additional and simple approach to utilizing data recipes.

Another problem is the limited scope of the search functionality. For example "ggd search variant" returns 35 packages, several of them from the ClinVar database, but "ggd search clinvar" returns no results. In a review of a more standard bioinformatics tool paper, the software details are less important, but the success of GGD will depend critically on its ability to build a community of recipe users and developers, and search functionality is a basic prerequisite for this.

We note that the reviewer correctly used the search function of GGD to search for different data packages and that specific packages related to ClinVar were not provided when searching for the "clinvar" term. As the reviewer mentioned, they correctly received 35 results for different data recipes that provide various types of variant information when searching for "variant" in GGD. 4 of the 35 resulting recipes are assembled by Ensembl and contain ClinVar associated variants. However, those 4 recipes do not show up when searching for "clinvar". The reason these 4 recipes do not show up is because of the complex search approach GGD uses to search and find various recipes. Simply, GGD will take each recipe and search the recipe's name and keywords for the search terms provided using a fuzzy matching search algorithm. Each recipe is given a match score and GGD uses a match score cutoff for inclusion/exclusion of search results. GGD also balances the size and proportion of search terms in order to not return too many results that are unrelated to the search term and relevant recipes. For example, if someone were to search for "re" looking for a reference genome and we did not set some criteria for filtering, this result could return 100's of recipes that are unrelated to the desired recipe.

Each of the 4 ClinVar associated recipes has a keyword “ClinVar-Associated”, however, because of restrictions imposed by GGD, this keyword gets a match score of 71, where GGD defaults to only displaying a recipe with a match score ≥ 75 . The match score can be adjusted using the “-m” parameter for ggd search, which would result in getting the 4 ClinVar associated variant recipes. However, we agree with the reviewer that the intended results from a search of “clinvar” should include these 4 recipes. Therefore, we have updated the GGD search approach to calculate a match score for all keywords and various subsets of keywords split using a set of predefined delimiters. We have also increased the match score default to 90. These changes reflect similar results when searching for various GGD recipes but also include recipes where the search term matches a keyword substring for a recipe. Additionally, the increased match score filters a similar number of recipes as before, restricting the results to only those that are highly similar to the search term. These changes are reflected in the latest version of GGD.

The features of the system are well-described, and because it is based on one of the most popular package managers, its commands and syntax are familiar to those with conda experience. The system runs smoothly and provides helpful feedback. One aspect that the paper could describe in greater detail is how a recipe is validated and what is required for it to pass continuous integration tests.

The aspects of validation and continuous integration were kept to a minimum in the manuscript because the manuscript is not a technical bioinformatics paper. We have added a supplemental note that covers several aspects of the CI system used by GGD. The manuscript now directs readers to the supplemental note and is highlighted in yellow.

While this paper is of interest to a more technical audience of bioinformaticians, it describes an approach to the problems of reproducible research and data management that is a novel application of technology, and, if made more accessible to less technical users and compatible with different omics data types, could be a substantial contribution. For this reason I recommend publication in a journal more specific to technical bioinformatics users.

Reviewers' Comments:

Reviewer #1:

Remarks to the Author:

I am pleased to report my concerns have been addressed. I hope to see this manuscript published soon!

Reviewer #2:

Remarks to the Author:

I read carefully the rebuttal of the authors. I understand they put some effort in answering to the issues raised by reviewers.

However, I am not convinced that GGD is a solution for reproducibility for genomic data. Main concern is still related on the long time support that would be required for a service like GGD.

As highlighted by another reviewer, I think the tool is of interest of a technical audience and should be published on a more technical journal.

Reviewer #3:

Remarks to the Author:

The authors have added text to place Go Get Data within the context of similar genomic tools, added a supplemental description of the continuous integration and validation system, and added a meta-recipe command that extends the base functionality of the system to support retrieval of datasets based on ID. These improve the description of the tool and its intended scope and satisfactorily address my concerns.